# Serum Kynurenic Acid and Kynurenine Are Negatively Associated with the Risk of Adult Moyamoya Disease

**DOI:** 10.3390/jcm11237069

**Published:** 2022-11-29

**Authors:** Xiaofan Yu, Peicong Ge, Yuanren Zhai, Wei Liu, Qian Zhang, Xun Ye, Xingju Liu, Rong Wang, Yan Zhang, Jizong Zhao, Dong Zhang

**Affiliations:** 1Department of Neurosurgery, Beijing Tiantan Hospital, Capital Medical University, Beijing 100069, China; 2China National Clinical Research Center for Neurological Diseases, Beijing 100061, China; 3Center of Stroke, Beijing Institute for Brain Disorders, Beijing 100069, China; 4Beijing Key Laboratory of Translational Medicine for Cerebrovascular Disease, Beijing 100069, China; 5Beijing Translational Engineering Center for 3D Printer in Clinical Neuroscience, Beijing 100069, China; 6Savaid Medical School, University of Chinese Academy of Sciences, Beijing 100043, China; 7Department of Neurosurgery, Beijing Hospital, Beijing 100020, China

**Keywords:** moyamoya disease, kynurenic acid, kynurenine, risk factors, biomarker

## Abstract

Background and aim. Kynurenine (KYN) and kynurenic acid (KYNA) are key intermediate metabolites associated with inflammation and immune responses in the kynurenine pathway. It remains unknown whether KYN or KYNA is associated with the risk of adult moyamoya disease (MMD). The aim of this study was to prospectively investigate the association between serum KYN or KYNA and the risk of adult MMD. Methods. The study was conducted from July 2020 to December 2021. We measured serum KYN and KYNA levels for 360 adult MMD patients (259 cases of ischemic MMD, 101 cases of hemorrhagic MMD) and 89 age-sex-matched healthy controls. Clinical and laboratory characteristics were collected from the medical record. Results. After multivariate adjustment, decreased serum KYNA (OR, 0.085; 95% CI, 0.035–0.206; *p* = 0.000) or KYN (OR, 0.430; 95% CI, 0.225–0.820; *p* = 0.010) levels were associated with increased risk of MMD when upper and lower tertiles were compared. In addition, a higher trend of hemorrhagic MMD was found in MMD patients in KYNA tertile 1 compared with those in tertile 2 to 3 (OR, 0.584; 95% CI, 0.345–0.987; *p* = 0.044). Addition of serum KYNA (net reclassification improvement: 73.24%, *p* = 0.000; integrated discrimination improvement: 9.60%, *p* = 0.000) or KYN (integrated discrimination improvement: 1.70%, *p* = 0.037) to conventional risk factors significantly improved the risk prediction of MMD. In the exploratory analysis, we observed an interaction between KYN and age (≥40 versus <40 years) or homocysteine levels (≥13.0 versus <13.0 μmol/L) on the risk of MMD. Conclusions. Decreased serum KYNA or KYN levels were associated with an increased risk of adult MMD, suggesting that serum KYNA or KYN may be a valuable predictive biomarker for adult MMD.

## 1. Introduction

Moyamoya disease (MMD) is an unusual type of cerebrovascular disorder associated with progressive stenosis of the intracranial internal carotid arteries and the aberrant vascular network in the brain [1,2]. In addition, MMD is the most common cause of stroke in children in China, Japan, and South Korea [3,4]. Although the epidemiology, clinical features, and treatment of MMD have been thoroughly studied [5,6], little is known about its etiology and progression. Many studies have indicated that inflammation and autoimmune responses may be important causes or progressive factors of MMD [7,8]. Fujimura et al. have discovered that MMD patients have considerably higher blood levels of CD163 and CXCL5 [7]. Weng et al. have found that circulating Treg and Th17 cells are considerably more prevalent in MMD patients [8]. However, the specific mechanism of inflammation or autoimmune responses associated with the onset of MMD remains unclear.

The main process for the degradation of tryptophan, known as the kynurenine pathway (KP), produces a number of biologically active metabolites. Kynurenine (KYN) and kynurenic acid (KYNA) are key intermediate metabolites associated with inflammation and immune responses in the KP [9]. While KYNA encourages monocyte extravasation and regulates cytokine release, KYN inhibits the action of natural killer cells, dendritic cells, or proliferating T cells [10]. In addition, KYN and KYNA have been linked to a number of diseases. Previous studies have suggested that KP is associated with the onset of tumor [11,12]. A number of studies have shown that KYN and KYNA are involved in the etiology of depression and schizophrenia [13,14]. Many clinical studies have shown that serum concentrations of KYN and KYNA are lower in patients with ischemic stroke [15,16]. Therefore, KYN or KYNA is expected to be the biomarker for immune or inflammatory diseases. However, the effects of circulating KYN or KYNA on the risk of MMD remained unclear.

The purpose of this study was to investigate the relationship between serum KYN or KYNA concentrations and the risk of adult MMD, hoping to provide new evidence for the pathogenesis of MMD.

## 2. Methods

### 2.1. Study Participants

On reasonable request, the corresponding author will provide the data that back up the study’s conclusions. From 1 July 2020 to 31 December 2021, we prospectively and continuously recruited adult MMD patients (18 years ≤ age ≤ 60 years) at the Department of Neurosurgery, Beijing Tiantan Hospital, Capital Medical University. All participants provided their written informed consent. The Ethics Committee of Beijing Tiantan Hospital gave its approval to the protocol of the study (Ethical inspection No. 2021YFC2500502).

MMD was identified by digital subtraction angiography, according to Japanese criteria released in 2012 [17]: (1) stenosis or blockage of the terminal internal carotid and the proximal middle and anterior cerebral arteries and (2) unilateral or bilateral involvement [18]. In total, 500 patients with MMD-type cerebrovascular disease (including 418 adult patients) from throughout the nation received care at our facility from 1 July 2020 to 31 December 2021. Among the adult patients, 11 patients were older than 60 years, 13 patients with moyamoya syndrome, and 34 patients without KYN and KYNA data were excluded. Moyamoya syndrome refers to patients with MMD-type cerebrovascular disease accompanied by other basic diseases, such as arteriosclerosis, autoimmune disease, meningitis, and Down syndrome [1]. Finally, 360 (86.12%) of the 418 patients were enrolled in the study (Appendix A). Control subjects were selected from healthy individuals who came for periodic health examinations aged similarly to the patients enrolled in this study. The periodic health examinations include routine blood tests, blood biochemical analysis, vital signs, height, and weight. None of these healthy participants or their close relatives had a history of systemic or cerebral vascular diseases, according to inquiries or their medical data.

### 2.2. Data Collection

After a 15-min break spent sitting, the participants’ right arms were examined for systolic blood pressure (SBP) and diastolic blood pressure (DBP) using a conventional mercury manometer. Electrocardiography data on heart rate (HR) was also captured. Weight (kg)/height (m^2^) was used to determine the body mass index (BMI). Peripheral blood samples were obtained in the morning after the participants had fasted for over 12 h. White blood cells (WBC), lymphocytes (LY), platelets (PLT), glucose (Glu), albumin (ALB), creatinine (Cr), uric acid (UA), triglyceride (TG), total cholesterol (TC), high-density lipoprotein cholesterol (HDL-C), low-density lipoprotein cholesterol (LDL-C), apolipoprotein A (apoA), apolipoprotein B (apoB), and homocysteine (Hcy) were all measured in fasting blood. The blood samples were collected before receiving treatment after admission.

Data on the MMD patients’ age, sex, and comorbidities, such as hypertension, coronary artery disease, smoking, drinking, diabetes mellitus, thyroid disorders, and initial clinical features (ischemia and hemorrhage), were gathered at the time of admission.

### 2.3. Quantification of KYNA and KYN

Serum levels of KYNA and KYN were quantified by ultra-performance liquid chromatography-tandem mass spectrometry (Sciex QTRAP 6500 LC-MS/MS). 50 μL of the serum sample was transferred to a centrifuge tube and mixed with 250 μL of 20% acetonitrile/methanol, vortexed for 3 min, centrifuged at 12,000 r/min for 10 min at 4 °C. Take 250 μL of supernatant into a new centrifuge tube and place the supernatant in −20 °C refrigerator for 30 min. Then the supernatant was centrifuged at 12,000 r/min for 10 min at 4 °C. After centrifugation, transfer 180 μL of the supernatant was through Protein Precipitation Plate for further LC-MS analysis.

### 2.4. Statistical Analysis

Baseline characteristics are presented as medians (interquartile range) for continuous variables and proportions for categorical variables. The Chi-square test and Mann–Whitney U test were used for categorical and continuous variables in baseline features, respectively.

The participants (360 MMD patients and 89 healthy controls) were divided into three categories according to the tertile of change in the KYNA or KYN index. The *p*-values for the trend were computed using the tertile of change in the KYNA or KYN index as the ordinal variable. The Cochran–Armitage test was used to assess trends in categorical variables, and one-way ANOVA was used to analyze continuous variables.

Odds ratios (ORs) of the risk of MMD (MMD overall, ischemic MMD, and hemorrhagic MMD) to serum levels of KYNA (continuous and categorical variables) or KYN (continuous and categorical variables) were assessed by the conditional logistic regression model, without and with modification for age, gender, HR, SBP, DBP, BMI, WBC, LY, PLT, Glu, Cr, UA, ALB, TG, TC, HDL-C, LDL-C, apoA, apoB, and Hcy. ORs of the risk of hemorrhagic MMD in the overall MMD cases to serum levels of KYNA (continuous and categorical variables) were evaluated by conditional logistic regression model [19], without and with modification for age, gender, WBC, PLT, TC, TG, and LDL-C. Possible modifications of the relationship between KYN levels as a binary variable (<502.53 ng/mL [tertile 1–2] versus ≥502.53 ng/mL [tertile 3]) and the risk of MMD (MMD overall, ischemic MMD, and hemorrhagic MMD) were assessed by subgroup analysis. Possible modifications of the relationship between KYNA levels as a binary variable (<7.65 ng/mL [tertile 1] versus ≥7.65 ng/mL [tertile 2–3]) and the risk of hemorrhagic MMD in the overall MMD cases were assessed by subgroup analysis.

Two statistical indices were used to measure the improvement in model performance resulting from the addition of new markers: net reclassification improvement and integrated discrimination improvement [20]. We created a conventional model (only including risk factors in model 2) and two novel models (including risk factors in model 2 and serum KYNA or KYN) by using logistic regression. We evaluated net reclassification improvement and integrated discrimination improvement by comparing the three models in order to determine whether adding serum KYNA or KYN to risk factors might enhance the prediction ability for the risk of MMD.

All statistical analyses were executed by using IBM SPSS Statistics (version 22.0; IBM Corp., Armonk, NY, USA) and R software (version 4.2.0; https://www.r-project.org; accessed on 1 October 2022). Statistical significance was regarded as *p* < 0.05 in a two-tailed test.

## 3. Results

### 3.1. Study Participants and Baseline Characteristics

In this study, we analyzed 360 MMD patients (Appendix A), comprising 259 cases of ischemic MMD and 101 cases of hemorrhagic MMD.

Table 1, Table 2 and Table 3 provide an overview of individuals’ baseline characteristics. Median values of serum KYNA or KYN levels in MMD and healthy participants were 8.55 ng/mL (inter-quartile range, 6.55–12.17) or 11.92 ng/mL (inter-quartile range, 9.64–17.15) and 472.50 ng/mL (inter-quartile range, 437.99–518.15) or 487.35 ng/mL (inter-quartile range, 448.30–539.11), respectively (Figure 1). Patients with MMD had lower KYNA (*p* = 0.000) or KYN (*p* = 0.035) levels than control subjects. Compared with control subjects, patients with MMD had lower concentrations of TC, LDL-C, HDL-C, and apoA (*p* < 0.05 for all). Furthermore, patients with MMD showed higher levels of SBP, DBP, BMI, WBC, TG, and Hcy (*p* < 0.05 for all) (Table 1). When all the participants were divided into tertiles according to their serum KYNA levels, participants with higher KYNA concentrations tended to be male and had higher BMI, LY, PLT, Glu, Cr, UA, LDL-C, apoB, Hcy, KYN levels; lower HDL-C and NLR levels; and a higher prevalence of smoking (*p* for trend < 0.05 for all) (Table 2). We stratified all participants into tertiles based on their serum KYN concentrations. Compared with participants with lower serum KYN levels, those with higher serum KYN levels were more likely to be male; had higher BMI, WBC, Cr, UA, TG, Hcy, and KYNA; lower HDL-C levels; and a higher prevalence of hypertension, smoking, and drinking (*p* for trend < 0.05 for all) (Table 3).

### 3.2. Association of KYNA Levels with the Risk of MMD

In general, after fully adjusting for age, gender, HR, SBP, DBP, BMI, WBC, LY, PLT, Glu, Cr, UA, ALB, TG, TC, HDL-C, LDL-C, apoA, apoB, and Hcy, the risk of MMD decreased by 23.4% with each increment KYNA level (per KYNA increment: OR, 0.766; 95% CI, 0.708–0.829) compared with healthy controls. When KYNA was evaluated as tertiles, compared with the lowest tertile (<7.65 ng/mL) of serum KYNA levels, the modified ORs (95% CI) for the risk of MMD in the middle (7.65 -< 11.55 ng/mL) and highest (≥11.55 ng/mL) tertiles were 0.132 (0.051–0.341) and 0.045 (0.016–0.123; *p* for trend = 0.000), respectively. When combined, the individuals in the upper tertiles (tertile 2–3 [≥7.65 ng/mL]: OR, 0.085; 95% CI, 0.035–0.206; *p* = 0.000) had a 91.50% lower risk of MMD than those in the lowest tertile (<7.65 ng/mL). Consistently, a significantly lower risk of ischemic MMD (OR, 0.075; 95% CI, 0.027–0.210) and a lower trend of hemorrhagic MMD (OR, 0.072; 95% CI, 0.024–0.219) was found in individuals in KYNA tertile 2 to 3 compared with those in tertile 1 (Table 4). In addition, we further analyzed the association between KYNA and the risk of hemorrhagic MMD in the overall MMD cases. After adjusting for age, gender, WBC, PLT, TC, TG, and LDL-C, a lower trend of hemorrhagic MMD (OR, 0.584; 95% CI, 0.345–0.987) were found in MMD patients in KYNA tertile 2 to 3 compared with those in tertile 1 (Appendix A).

### 3.3. Association of KYN Levels with the Risk of MMD

Similarly, after adjusting for age, gender, HR, SBP, DBP, BMI, WBC, LY, PLT, Glu, Cr, UA, ALB, TG, TC, HDL-C, LDL-C, apoA, apoB, and Hcy, the risk of MMD decreased by 0.7% with each increment KYN level (per KYN increment: OR, 0.993; 95% CI, 0.988–0.997) compared with healthy controls on continuous analysis. After KYN was stratified as tertiles, compared with the lowest tertile (<451.68 ng/mL) of serum KYN concentrations, the modified ORs (95% CI) for the risk of MMD in the middle (451.68-<502.53 ng/mL) and highest (≥502.53 ng/mL) tertiles were 0.906 (0.426–1.926) and 0.408 (0.191–0.871; *p* for trend = 0.020), respectively. After combination, the individuals in the highest tertile (tertile 3 [≥502.53 ng/mL]: OR, 0.430; 95% CI, 0.225–0.820; *p* = 0.010) had a 57.00% lower risk of MMD than those in the lower tertiles (<502.53 ng/mL). Consistently, a significantly lower risk of ischemic MMD (OR, 0.387; 95% CI, 0.184–0.814) and a lower trend of hemorrhagic MMD (OR, 0.429; 95% CI, 0.187–0.986) were found in individuals in KYN tertile 3 compared with those in tertile 1 to 2 (Table 5).

### 3.4. Stratified Analysis by Important Covariables

The relationship between serum KYN levels (≥502.53 ng/mL versus <502.53 ng/mL) and the risk of overall MMD, ischemic MMD, and hemorrhagic MMD in different subgroups was evaluated using stratified analyses (Appendix A). We observed interactions between age (≥40 versus <40 years) or Hcy levels (≥13.0 versus <13.0 μmol/L) and serum KYN on the risk of overall MMD (*p* for interaction, 0.042 or 0.025; Appendix A) or ischemic MMD (*p* for interaction, 0.047 or 0.023; Appendix A) but not for hemorrhagic MMD (*p* for interaction, 0.132 or 0.067; Appendix A). Furthermore, the stratified analyses of the association between KYNA and the risk of hemorrhagic MMD in the overall MMD cases did not show any differences (Appendix A).

### 3.5. Incremental Predictive Value of Serum KYNA or KYN for the Risk of MMD

We examined whether adding serum KYNA or KYN to the conventional model (all risk factors in model 2) could enhance its ability to predict the risk of overall MMD, ischemic MMD, and hemorrhagic MMD. The addition of KYNA tertile to conventional risk factors significantly improved the risk reclassification for the risk of overall MMD (net reclassification improvement: 73.20%, *p* = 0.000; integrated discrimination improvement: 9.60%, *p* = 0.000), ischemic MMD (net reclassification improvement: 82.10%, *p* = 0.000; integrated discrimination improvement: 8.70%, *p* = 0.000), and hemorrhagic MMD (net reclassification improvement: 116.00%, *p* = 0.000; integrated discrimination improvement: 31.40%, *p* = 0.000). In addition, adding KYN tertile to the model resulted in an integrated discrimination improvement of 1.7% (*p* = 0.037 or *p* = 0.043) for both overall MMD and ischemic MMD (Appendix A).

## 4. Discussion

To our knowledge, this is the first clinical study to examine the prospective association between serum KYN or KYNA and the risk of adult MMD. In the present study, we found that decreased serum KYN or KYNA levels were independently associated with an increased risk of adult MMD (MMD overall, ischemic MMD, and hemorrhagic MMD). Adding serum KYNA or KYN tertile to conventional risk factors (the risk factors in model 2) could improve risk prediction for the risk of MMD (MMD overall, ischemic MMD, and hemorrhagic MMD for KYNA tertile; MMD overall and ischemic MMD for KYN tertile). In addition, decreased serum KYNA levels were independently related to the risk of hemorrhagic MMD in the overall MMD cases. These findings suggested that serum KYNA or KYN might be a potential biomarker for the risk of MMD, which could bring additional diagnostic information to physicians.

It has been reported that KYNA and KYN have anti-inflammatory properties. Tryptophan is an essential amino acid critical for protein synthesis. As the major route of tryptophan catabolism, KP metabolism leads to a number of neuroactive metabolites, including KYNA and KYN. Tryptophan is metabolized by the KP’s rate-limiting enzymes tryptophan 2, 3-dioxygenase (TDO) and indoleamine 2, 3-dioxygenase (IDO) into KYN. KYNA is produced from KYN by the kynurenine aminotransferases (KATs) [10]. KYN reduces the action of natural killer cells and dendritic cells, whereas KYNA controls cytokine release [10,21]. Furthermore, it has been reported that KYNA can stimulate lipid metabolism and improve Glu tolerance [22]. Growing evidence has indicated that the KP is crucial in the pathophysiology of stroke [15,16]. Mo et al. enrolled 81 patients with ischemic stroke and 35 healthy controls and concluded a decrease in serum levels of KYNA in the stroke group [16]. Hajsl et al. have studied 18 patients with acute ischemic stroke and 25 healthy controls and reported a decrease in serum levels of KYN in the stroke group [15].

Although there are few population-based data on the association between KYN or KYNA and the risk of adult MMD, our findings were similar to those of previous studies on stroke. In this study, adult patients with MMD had lower serum KYN and KYNA concentrations compared with healthy controls. Furthermore, decreased serum KYN and KYNA levels were independently associated with an increased risk of adult MMD in the regression analyses. So, it is reasonable to speculate that the inhibition of KP (downregulation of IDO or TDO) may be associated with the risk of adult MMD. A decreased expression of IDO or TDO may contribute to lower levels of KYN and KYNA. The relationship between the rate-limiting enzymes of KP and the pathogenesis of MMD needs to be confirmed by further basic research. Increased BMI and Hcy were linked to a higher risk of adult MMD according to our earlier case-control study. However, a decreased incidence of adult MMD was associated with higher HDL-C levels [23]. In this study, we also found that adult patients with MMD had higher levels of BMI and Hcy and lower levels of HDL-C compared with control subjects. Interestingly, although patients with MMD had lower KYNA and KYN levels than control subjects, BMI and Hcy levels were positively correlated with KYNA or KYN levels; HDL-C levels were negatively correlated with KYNA or KYN. This suggested that although Hcy and BMI levels decreased with lower KYNA or KYN levels in MMD patients, Hcy and BMI levels in MMD patients were generally higher than those in control subjects.

Another intriguing result of the current investigation is the modification of the effect of KYN on the incidence of both overall MMD and ischemic MMD by age and Hcy concentrations. As shown in Appendix A, those with high age and KYN or low Hcy and high KYN had the lowest rate of overall MMD and ischemic MMD. If further identified, maintaining both high KYN and low Hcy levels would be an available MMD prevention tactic.

Clinical signs and characteristics, such as collateral vasculature, blood-brain barrier integrity, and angiogenesis potentials, differ amongst MMD subtypes [24,25,26]. However, uncertainty persists on the pathophysiology that underlies MMD phenotypes. Our study demonstrated that decreased serum KYNA levels were independently related to the risk of hemorrhagic MMD in the overall MMD cases. It is unclear how decreased KYNA affects hemorrhagic MMD. The results of an in vitro study have indicated that KYNA could contribute to maintaining or restoring the protective function of the endothelium. So, we speculated that decreased KYNA might lead to vascular endothelial injury. The relationship between KYNA and hemorrhagic MMD needs to be further investigated in depth.

For a better knowledge of the pathophysiology of MMD, our findings have significant clinical ramifications. However, this study has a number of limitations. First, this study was observational; the number of participants was not equal between the groups. Thus, even though we had practically all of the significant confounders corrected, the possibility of residual confounding could not be completely ruled out. Second, because only Chinese adult patients with MMD were included, these results could not be applied to children or other races. Third, we only measured preoperative KYN or KYNA concentrations during hospitalization; this prevented us from studying the relationship between dynamic variations in serum KYN or KYNA and the prognosis of MMD. Fourth, we did not consider the severity of MMD when we investigated the relationship of KYN or KYNA with the risk of MMD. Fifth, patients with other intracranial vascular diseases, such as atherosclerosis, were not included in this study as a control group. Finally, this is a prospective study using clinical data from a single center. The results’ applicability is constrained, necessitating additional validation in a different cohort.

## 5. Conclusions

In conclusion, our study indicated a significant association of decreased KYN or KYNA levels with the risk of adult MMD. The addition of serum KYN or KYNA to conventional risk factors could improve the risk prediction of adult MMD, suggesting that serum KYN or KYNA might be a potential predictive biomarker for adult MMD. Furthermore, serum KYNA is an independent risk factor of hemorrhagic MMD and may function as a biomarker for hemorrhage prediction in MMD.

## Figures and Tables

**Figure 1 jcm-11-07069-f001:**
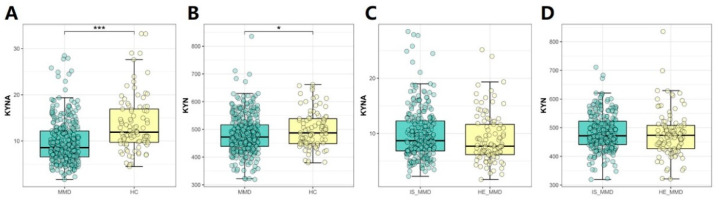
Comparison of serum KYNA and KYN levels between MMD patients/ischemic MMD and healthy controls/hemorrhagic MMD. (**A**,**B**) Serum KYNA and KYN levels were significantly different between MMD patients and healthy controls. *** *p* < 0.001; * 0.01 < *p* < 0.05. (**C**,**D**) Serum KYNA and KYN levels did not show differences between ischemic MMD and hemorrhagic MMD. MMD indicates moyamoya disease; HC, healthy controls; KYNA, kynurenic acid; KYN, kynurenine; IS_MMD, ischemic moyamoya disease; HE_MMD, hemorrhagic moyamoya disease. The unit of measurement is ng/mL.

**Table 1 jcm-11-07069-t001:** Population characteristics among controls and MMD cases.

Characteristics Analyzed	MMD (*n*)		*p*
	Yes (360)	No (89)	
Age, y	43.00 (34.00–49.00)	39.00 (31.00–50.00)	0.209
Men (%)	150 (41.67)	37 (41.57)	0.987
Heart rate, bpm	78.00 (75.00–80.00)	78.00 (70.00–85.00)	0.114
SBP, mmHg	132.00 (124.00–140.00)	124.00 (115.50–130.00)	0
DBP, mmHg	81.00 (76.00–89.00)	78.00 (74.00–82.00)	0.001
BMI, kg/m^2^	25.00 (22.49–27.78)	23.51 (21.41–26.47)	0.007
Medical history (%)			
Hypertension	131 (36.89)	0 (0)	0
Diabetes	59 (16.39)	0 (0)	0
Coronary artery disease	7 (1.94)	0 (0)	0.185
Hyperlipidemia	54 (15.00)	0 (0)	0
Thyroid disease	21 (5.83)	0 (0)	0.02
Smoking	71 (19.72)	2 (2.25)	0
Drinking	42 (11.67)	0 (0)	0.001
Laboratory results, median (IQR)			
WBC count, 10^9^/L	6.81 (5.64–8.12)	6.03 (5.01–6.89)	0
Lymphocyte count, 10^9^/L	1.92 (1.54–2.41)	1.91 (1.52–2.23)	0.247
Platelet count, 10^9^/L	248.00 (209.50–284.75)	233.00 (201.50–288.50)	0.271
Glucose, mmol/L	5.11 (4.71–5.77)	5.04 (4.75–5.37)	0.229
Triglyceride, mmol/L	1.20 (0.82–1.64)	0.87 (0.65–1.26)	0
Total cholesterol, mmol/L	4.23 (3.55–4.83)	4.62 (4.08–5.06)	0
HDL-C, mmol/L	1.30 (1.12–1.49)	1.53 (1.33–1.74)	0
LDL-C, mmol/L	2.40 (1.84–2.98)	2.69 (2.26–3.13)	0.001
apoA, g/L	1.29 (1.15–1.45)	1.39 (1.26–1.54)	0
apoB, g/L	0.82 (0.70–0.97)	0.77 (0.69–0.95)	0.329
Homocysteine	12.00 (9.30–15.08)	10.62 (8.67–12.63)	0.001
Kynurenic acid, ng/mL	8.55 (6.55–12.17)	11.92 (9.64–17.15)	0
Kynurenine, ng/mL	472.50 (437.99–518.15)	487.35 (448.30–539.11)	0.035

SBP indicates systolic blood pressure; DBP, diastolic blood pressure; BMI, body mass index; WBC, white blood cells; HDL-C, high-density lipoprotein cholesterol; LDL-C, low-density lipoprotein cholesterol; apoA, apolipoprotein A; apoB, apolipoprotein B; MMD, moyamoya disease; IQR, interquartile range.

**Table 2 jcm-11-07069-t002:** Baseline characteristics of participants according to tertiles of changes in kynurenic acid.

Characteristics	Tertiles of Changes in Kynurenic Acid			*p* for Trend
	T1 (<7.65)	T2 (7.65 -< 11.55)	T3 (≥11.55)	
No. of participants	149	150	150	
Age, years	40.00 (33.00–47.00)	45.00 (34.00–50.25)	43.00 (33.00–51.00)	0.097
Men (%)	41 (27.52)	62 (41.33)	84 (56.00)	0.000
Medical history (%)				
Hypertension	34 (22.82)	49 (32.67)	48 (32.00)	0.081
Diabetes	19 (12.75)	22 (14.67)	18 (12.00)	0.846
Coronary artery disease	1 (0.67)	2 (1.33)	4 (2.67)	0.164
Hyperlipidemia	9 (6.04)	27 (18.00)	18 (12.00)	0.114
Thyroid disease	12 (8.05)	4 (2.67)	5 (3.33)	0.054
Smoking	11 (7.38)	31 (20.67)	31 (20.67)	0.002
Drinking	7 (4.70)	19 (12.67)	16 (10.67)	0.077
Clinical feature				
Heart rate, bpm	78.00 (75.00–80.00)	78.00 (72.00–80.25)	78.00 (73.75–81.25)	0.757
SBP, mmHg	130.00 (120.00–138.00)	130.00 (123.00–140.00)	130.00 (124.00–140.00)	0.265
DBP, mmHg	78.00 (74.00–87.00)	81.00 (77.00–89.00)	80.00 (75.00–88.25)	0.120
BMI, kg/m^2^	23.95 (21.59–26.23)	24.92 (22.17–27.82)	25.40 (23.12–28.03)	0.003
Laboratory results, median (IQR)				
WBC count, 10^9^/L	6.56 (5.60–7.73)	6.78 (5.44–7.72)	6.56 (5.56–8.15)	0.958
Lymphocyte count, 10^9^/L	1.78 (1.43–2.26)	1.93 (1.52–2.32)	2.01 (1.59–2.48)	0.018
Platelet count, 10^9^/L	256.00 (217.00–296.00)	243.50 (203.25–279.50)	243.00 (204.75–271.25)	0.010
Glucose, mmol/L	5.05 (4.66–5.76)	5.14 (4.69–5.71)	5.09 (4.75–5.48)	0.017
Creatinine, μmol/L	50.30 (44.90–60.75)	55.15 (46.43–65.83)	63.75 (52.70–71.23)	0.000
Uric acid, μmol/L	279.60 (226.20–331.35)	302.90 (249.08–362.95)	348.05 (292.23–408.58)	0.000
Albumin, g/L	45.70 (43.60–47.60)	45.15 (43.40–47.13)	45.15 (43.28–46.73)	0.358
Triglyceride, mmol/L	0.91 (0.72–1.31)	1.25 (0.84–1.82)	1.23 (0.82–1.65)	0.248
Total cholesterol, mmol/L	4.21 (3.52–4.72)	4.24 (3.62–4.91)	4.41 (3.83–5.00)	0.058
HDL-C, mmol/L	1.35 (1.18–1.58)	1.33 (1.15–1.57)	1.31 (1.13–1.47)	0.027
LDL-C, mmol/L	2.34 (1.80–2.86)	2.40 (1.91–2.95)	2.56 (2.09–3.14)	0.009
apoA, g/L	1.33 (1.17–1.47)	1.32 (1.18–1.50)	1.29 (1.15–1.44)	0.287
apoB, g/L	0.77 (0.66–0.93)	0.82 (0.70–0.98)	0.86 (0.72–0.99)	0.006
Homocysteine, μmol/L	10.60 (8.65–14.15)	11.28 (9.20–13.71)	12.15 (9.59–15.70)	0.001
Kynurenine, ng/mL	441.00 (412.25–470.01)	475.13 (450.66–504.19)	535.67 (488.26–579.47)	0.000

SBP indicates systolic blood pressure; DBP, diastolic blood pressure; BMI, body mass index; WBC, white blood cells; HDL-C, high-density lipoprotein cholesterol; LDL-C, low-density lipoprotein cholesterol; apoA, apolipoprotein A; apoB, apolipoprotein B; IQR, interquartile range.

**Table 3 jcm-11-07069-t003:** Baseline characteristics of participants according to tertiles of changes in kynurenine.

Characteristics	Tertiles of Changes in Kynurenine			*p* for Trend
	T1 (<451.68)	T2 (451.68 -< 502.53)	T3 (≥502.53)	
No. of participants	149	150	150	
Age, years	42.00 (34.00–47.00)	41.00 (31.75–49.25)	44.00 (34.00–51.25)	0.176
Men (%)	41 (27.52)	70 (46.67)	76 (50.67)	0.000
Medical history (%)				
Hypertension	36 (24.16)	39 (26.00)	56 (37.33)	0.012
Diabetes	23 (15.44)	16 (10.67)	20 (13.33)	0.000
Coronary artery disease	1 (0.67)	2 (1.33)	4 (2.67)	0.164
Hyperlipidemia	14 (9.40)	15 (10.00)	25 (16.67)	0.053
Thyroid disease	6 (4.03)	8 (5.33)	7 (4.67)	0.794
Smoking	12 (8.05)	33 (22.00)	28 (18.67)	0.013
Drinking	4 (2.68)	22 (14.67)	16 (10.67)	0.018
Clinical feature				
Heart rate, bpm	78.00 (72.00–80.00)	78.50 (76.00–82.50)	78.00 (74.00–80.25)	0.008
SBP, mmHg	130.00 (120.00–140.00)	130.00 (122.00–138.00)	132.00 (124.00–140.00)	0.060
DBP, mmHg	80.00 (76.00–88.00)	80.00 (75.00–88.00)	80.00 (75.00–88.25)	0.247
BMI, kg/m^2^	23.88 (21.82–26.03)	25.28 (22.04–27.80)	25.40 (23.05–28.34)	0.000
Laboratory results, median (IQR)				
WBC count, 10^9^/L	6.42 (5.37–7.65)	6.79 (5.62–8.24)	6.71 (5.73–8.07)	0.036
Lymphocyte count, 10^9^/L	1.82 (1.48–2.27)	1.93 (1.56–2.33)	1.98 (1.55–2.47)	0.071
PLT, 10^9^/L	257.00 (215.00–297.50)	236.00 (202.75–276.75)	245.00 (206.75–279.00)	0.132
Glucose, mmol/L	5.05 (4.64–5.81)	5.06 (4.67–5.57)	5.12 (4.77–5.53)	0.777
Creatinine, μmol/L	50.90 (45.15–60.65)	55.90 (47.38–66.90)	61.35 (49.90–70.90)	0.000
Uric acid, μmol/L	272.30 (226.85–326.40)	312.40 (262.30–379.10)	346.65 (276.73–405.95)	0.000
Albumin, g/L	45.50 (43.50–47.35)	45.20 (43.40–47.30)	45.30 (43.40–47.00)	0.597
Triglyceride, mmol/L	0.90 (0.70–1.48)	1.21 (0.82–1.67)	1.24 (0.91–1.57)	0.002
Total cholesterol, mmol/L	4.23 (3.65–4.83)	4.28 (3.66–4.99)	4.27 (3.71–4.87)	0.335
HDL-C, mmol/L	1.35 (1.19–1.59)	1.32 (1.15–1.53)	1.32 (1.13–1.48)	0.016
LDL-C, mmol/L	2.41 (1.87–2.96)	2.40 (2.01–3.05)	2.42 (1.93–3.14)	0.298
apoA, g/L	1.33 (1.19–1.49)	1.30 (1.16–1.44)	1.30 (1.15–1.47)	0.147
apoB, g/L	0.77 (0.68–0.95)	0.82 (0.68–0.97)	0.82 (0.71–0.99)	0.054
Homocysteine, μmol/L	10.82 (8.69–14.15)	11.59 (9.29–13.53)	12.01 (9.60–15.74)	0.004
kynurenic acid, ng/mL	7.14 (5.18–9.13)	8.60 (6.91–10.81)	13.23 (10.18–17.49)	0.000

SBP indicates systolic blood pressure; DBP, diastolic blood pressure; BMI, body mass index; WBC, white blood cells; HDL-C, high-density lipoprotein cholesterol; LDL-C, low-density lipoprotein cholesterol; apoA, apolipoprotein A; apoB, apolipoprotein B; IQR, interquartile range.

**Table 4 jcm-11-07069-t004:** The association between kynurenic acid and the risk of MMD.

Kynurenic Acid, ng/mL	No. of Events (%)	Crude		Model 1 *		Model 2 †	
		OR (95% CI)	*p* Value	OR (95% CI)	*p* Value	OR (95% CI)	*p* Value
MMD overall							
Continuous	360 (80.18)	0.874 (0.836–0.914)	0.000	0.830 (0.787–0.876)	0.000	0.766 (0.708–0.829)	0.000
Categories							
Tertiles							
T1 (<7.65)	137 (91.95)	1.0 (Ref)		1.0 (Ref)		1.0 (Ref)	
T2 (7.65 -< 11.55)	120 (80.00)	0.350 (0.172–0.715)	0.004	0.236 (0.109–0.509)	0.000	0.132 (0.051–0.341)	0.000
T3 (≥11.55)	103 (68.67)	0.192 (0.097–0.380)	0.000	0.106 (0.049–0.231)	0.000	0.045 (0.016–0.123)	0.000
*p* for trend		0.000		0.000		0.000	
T1 (<7.65)	137 (91.95)	1.0 (Ref)		1.0 (Ref)		1.0 (Ref)	
T2–3 (≥7.65)	223 (74.33)	0.254 (0.133–0.483)	0.000	0.162 (0.080–0.328)	0.000	0.085 (0.035–0.206)	0.000
Ischemic MMD							
Continuous	259 (71.94)	0.879 (0.838–0.922)	0.000	0.834 (0.786–0.884)	0.000	0.764 (0.697–0.837)	0.000
Categories							
Tertiles							
T1 (<7.65)	90 (88.24)	1.0 (Ref)		1.0 (Ref)		1.0 (Ref)	
T2 (7.65 -< 11.55)	93 (75.61)	0.413 (0.199–0.857)	0.018	0.273 (0.121–0.616)	0.002	0.116 (0.039–0.346)	0.000
T3 (≥11.55)	76 (61.79)	0.216 (0.107–0.436)	0.000	0.113 (0.049–0.261)	0.000	0.036 (0.011–0.122)	0.000
*p* for trend		0.000		0.000		0.000	
T1 (<7.65)	90 (88.24)	1.0 (Ref)		1.0 (Ref)		1.0 (Ref)	
T2–3 (≥7.65)	169 (68.70)	0.293 (0.151–0.566)	0.000	0.182 (0.086–0.386)	0.000	0.075 (0.027–0.210)	0.000
Hemorrhagic MMD							
Continuous	101 (28.06)	0.839 (0.783–0.900)	0.000	0.801 (0.738–0.870)	0.000	0.693 (0.607–0.792)	0.000
Categories							
Tertiles							
T1 (<7.65)	47 (79.66)	1.0 (Ref)		1.0 (Ref)		1.0 (Ref)	
T2 (7.65 -< 11.55)	27 (47.37)	0.230 (0.101–0.522)	0.000	0.173 (0.069–0.430)	0.000	0.126 (0.039–0.408)	0.001
T3 (≥11.55)	27 (36.49)	0.147 (0.066–0.324)	0.000	0.097 (0.039–0.238)	0.000	0.034 (0.009–0.124)	0.000
*p* for trend		0.000		0.000		0.000	
T1 (<7.65)	47 (79.66)	1.0 (Ref)		1.0 (Ref)		1.0 (Ref)	
T2–3 (≥7.65)	54 (41.22)	0.179 (0.087–0.369)	0.000	0.126 (0.056–0.287)	0.000	0.072 (0.024–0.219)	0.000

MMD indicates moyamoya disease; OR, odds ratio. * Model 1 was adjusted for age, gender, heart rate, SBP, DBP, and BMI. † Model 2 was adjusted for all the variables in model 1 plus WBC count, lymphocyte count, platelet count, glucose, creatinine, uric acid, albumin, triglyceride, total cholesterol, HDL-C, LDL-C, apoA, ApoB, and homocysteine.

**Table 5 jcm-11-07069-t005:** The association between kynurenine and the risk of MMD.

Kynurenine, ng/mL	No. of Events (%)	Crude		Model 1 *		Model 2 †	
		OR (95% CI)	*p* Value	OR (95% CI)	*p* Value	OR (95% CI)	*p* Value
MMD overall							
Continuous	360 (80.18)	0.996 (0.993–1.000)	0.030	0.995 (0.991–0.998)	0.004	0.993 (0.988–0.997)	0.002
Categories							
Tertiles							
T1 (<451.68)	122 (81.88)	1.0 (Ref)		1.0 (Ref)		1.0 (Ref)	
T2 (451.68 -< 502.53)	125 (83.33)	1.107 (0.608–2.013)	0.740	1.035 (0.544–1.969)	0.915	0.906 (0.426–1.926)	0.797
T3 (≥502.53)	113 (75.33)	0.676 (0.387–1.181)	0.169	0.497 (0.267–0.926)	0.028	0.408 (0.191–0.871)	0.021
*p* for trend		0.156		0.026		0.020	
T1–2 (<502.53)	122 (81.88)	1.0 (Ref)		1.0 (Ref)		1.0 (Ref)	
T3 (≥502.53)	238 (79.33)	0.643 (0.399–1.036)	0.069	0.488 (0.287–0.830)	0.008	0.430 (0.225–0.820)	0.010
Ischemic MMD							
Continuous	259 (71.94)	0.996 (0.992–1.000)	0.036	0.994 (0.989–0.998)	0.004	0.991 (0.985–0.997)	0.003
Categories							
Tertiles							
T1 (<451.68)	56 (49.56)	1.0 (Ref)		1.0 (Ref)		1.0 (Ref)	
T2 (451.68 -< 502.53)	89 (78.07)	1.118 (0.602–2.077)	0.725	1.045 (0.526–2.080)	0.899	1.047 (0.444–2.473)	0.916
T3 (≥502.53)	84 (69.42)	0.713 (0.399–1.273)	0.253	0.483 (0.247–0.946)	0.034	0.397 (0.164–0.964)	0.041
*p* for trend		0.234		0.031		0.036	
T1–2 (<502.53)	145 (63.88)	1.0 (Ref)		1.0 (Ref)		1.0 (Ref)	
T3 (≥502.53)	84 (69.42)	0.675 (0.411–1.107)	0.119	0.473 (0.267–0.838)	0.010	0.387 (0.184–0.814)	0.012
Hemorrhagic MMD							
Continuous	101 (28.06)	0.996 (0.992–1.000)	0.068	0.995 (0.991–1.000)	0.040	0.993 (0.987–0.999)	0.022
Categories							
Tertiles							
T1 (<451.68)	36 (57.14)	1.0 (Ref)		1.0 (Ref)		1.0 (Ref)	
T2 (451.68 -< 502.53)	36 (59.02)	1.080 (0.529–2.205)	0.833	1.112 (0.513–2.410)	0.787	1.031 (0.408–2.610)	0.948
T3 (≥502.53)	29 (43.94)	0.588 (0.293–1.180)	0.135	0.499 (0.231–1.079)	0.077	0.436 (0.165–1.150)	0.093
*p* for trend		0.130		0.073		0.093	
T1–2 (<502.53)	72 (58.06)	1.0 (Ref)		1.0 (Ref)		1.0 (Ref)	
T3 (≥502.53)	29 (43.94)	0.566 (0.310–1.034)	0.064	0.473 (0.243–0.918)	0.027	0.429 (0.187–0.986)	0.046

MMD indicates moyamoya disease; OR, odds ratio. * Model 1 was adjusted for age, gender, heart rate, SBP, DBP, and BMI. † Model 2 was adjusted for all the variables in model 1 plus WBC count, lymphocyte count, platelet count, glucose, creatinine, uric acid, albumin, triglyceride, total cholesterol, HDL-C, LDL-C, apoA, apoB, and homocysteine.

## Data Availability

The data that support the findings of this study are available from the corresponding authors on reasonable request.

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
