# Peer review of "Serum Kynurenic Acid and Kynurenine Are Negatively Associated with the Risk of Adult Moyamoya Disease"

_jcm, 2022, doi:10.3390/jcm11237069_

Round 1

Reviewer 1 Report

The authors measured the serum levels of Kynurenine and kynurenic acid in patients with moyamoya disease, which are involved in inflammation and immune responses. The authors showed that their reduction is involved in moyamoya disease. I believe it is a useful report with high novelty and adequate sample size.

I think the data will be better understood by adding box-and-whisker plots for Kynurenine and kynurenic acid levels in moyamoya disease and control, and also for hemorrhagic and ischemic type moyamoya disease. Please consider.

Author Response

Dear Reviewer 1:

On behalf of my co-authors, we thank you very much for giving us an opportunity to revise our article entitled Serum kynurenic acid and kynurenine are negatively associated with the risk of adult moyamoya disease” (jcm-2025866). We appreciate you very much for your positive and constructive comments and suggestions on this article. We have studied your comments carefully and have made revision which marked as red in the manuscript. We have tried our best to revise our manuscript according to the comments, which we hope will meet with approval. We would like to express our great appreciation to you for spending the time and effect on our article.

The main corrections and the responses to reviewers’ comments are as follows:

Reviewers' comments:

Reviewer #1

Comment 1: I think the data will be better understood by adding box-and-whisker plots for Kynurenine and kynurenic acid levels in moyamoya disease and control, and also for hemorrhagic and ischemic type moyamoya disease. Please consider.

Response: Thank you for your comments. We have added the box-and-whisker plots for kynurenine and kynurenic acid levels in MMD and control, and also for hemorrhagic and ischemic type MMD according to your requirement. You can see them in Figure 1.

Once again, we would like to express our great appreciation to you for comments on our article. We hope our corrections will meet with your approval. Looking forward to hearing from you soon.

Best Regards.

Yours Sincerely,

Dong Zhang

Reviewer 2 Report

This well written paper reports that decreased serum KYN or KYNA levels were independently associated with increased risk of MMD in adult patients.

The methodology seems to be sound and relevant references are included.

The main problem concerns the lack of a control group with patients suffering from other (‘regular’) intracranial vascular diseases such as atherosclerosis. Therefore, we do not know if the associations as described could be (also) the result of the vascular damage in the brain, whereas there is no association with MMD vascular pathology.

Another minor problem is the poor description of the control subjects. They were selected from healthy individuals who came for regular examinations, but for what type of regular examinations these controls came to the hospital? Did they suffer from vascular diseases?

Author Response

Dear Reviewer 2:

On behalf of my co-authors, we thank you very much for giving us an opportunity to revise our article entitled Serum kynurenic acid and kynurenine are negatively associated with the risk of adult moyamoya disease” (jcm-2025866). We appreciate you very much for your positive and constructive comments and suggestions on this article. We have studied your comments carefully and have made revision which marked as red in the manuscript. We have tried our best to revise our manuscript according to the comments, which we hope will meet with approval. We would like to express our great appreciation to you for spending the time and effect on our article.

The main corrections and the responses to reviewers’ comments are as follows:

Reviewers' comments:

Reviewer #2

Comment 1: The main problem concerns the lack of a control group with patients suffering from other (‘regular’) intracranial vascular diseases such as atherosclerosis. Therefore, we do not know if the associations as described could be (also) the result of the vascular damage in the brain, whereas there is no association with MMD vascular pathology.

Response: Thank you for your comments. Your suggestions are constructive and helpful. However, the present study is a sex-age matched case-control study. Patients with MMD have a younger age of onset than patients with atherosclerosis. So, on this point, these two groups of patients are not particularly comparable. We will focus more on this issue in our future research efforts. We have added the sentence “Fifth, patients with other intracranial vascular diseases, such as atherosclerosis, were not included in this study as a control group.” in the Discussion section (page 11, line 317-318).

Comment 2: Another minor problem is the poor description of the control subjects. They were selected from healthy individuals who came for regular examinations, but for what type of regular examinations these controls came to the hospital? Did they suffer from vascular diseases?

Response: Thank you for your comments. Control subjects were selected from healthy individuals who came for periodic health examinations. The periodic health examinations include routine blood tests, blood biochemical analysis, vital signs, height, and weight. None of these healthy participants or their close relatives had a history of systemic or cerebral vascular diseases, according to inquiries or their medical data.

  We have revised the sentence “Control subjects were selected from healthy individuals who came for regular examinations aged similarly to the patients enrolled in this study.” to “Control subjects were selected from healthy individuals who came for periodic health examinations aged similarly to the patients enrolled in this study.” (page 2, line 83-85); we have added the sentence “The periodic health examinations include routine blood tests, blood biochemical analysis, vital signs, height, and weight.” in the Methods section (page 2, line 85-86); we have revised the sentence “None of these healthy participants or their close relatives had a history of MMD or cardiac disorders, according to inquiries or their medical data.” to “None of these healthy participants or their close relatives had a history of systemic or cerebral vascular diseases, according to inquiries or their medical data.” (page 2, line 86-88).

Once again, we would like to express our great appreciation to you for comments on our article. We hope our corrections will meet with your approval. Looking forward to hearing from you soon.

Best Regards.

Yours Sincerely,

Dong Zhang